# Age-seroprevalence curves for the multi-strain structure of influenza A virus

Dao Nguyen Vinh[1], Nguyen Thi Duy Nhat[1,2,3], Erwin de Bruin[4], Nguyen Ha Thao Vy[1], Tran Thi Nhu Thao[1], Huynh Thi Phuong [1], Pham Hong Anh[1], Stacy Todd[1,5,6], Tran Minh Quan [1], Nguyen Thi Le Thanh[1], Nguyen Thi Nam Lien[7], Nguyen Thi Hong Ha[8], Tran Thi Kim Hong[9], Pham Quang Thai [10], Marc Choisy[1,2], Tran Dang Nguyen[3], Cameron P. Simmons[11], Guy E. Thwaites [1,2], Hannah E. Clapham [1,2,12], Nguyen Van Vinh Chau [13], Marion Koopmans [4] & Maciej F. Boni [1,2,3✉]

The relationship between age and seroprevalence can be used to estimate the annual attack rate of an infectious disease. For pathogens with multiple serologically distinct strains, there is a need to describe composite exposure to an antigenically variable group of pathogens. In this study, we assay 24,402 general-population serum samples, collected in Vietnam between 2009 to 2015, for antibodies to eleven human influenza A strains. We report that a principal components decomposition of antibody titer data gives the first principal component as an appropriate surrogate for seroprevalence; this results in annual attack rate estimates of 25.6% (95% CI: 24.1% – 27.1%) for subtype H3 and 16.0% (95% CI: 14.7% – 17.3%) for subtype H1. The remaining principal components separate the strains by serological similarity and associate birth cohorts with their particular influenza histories. Our work shows that dimensionality reduction can be used on human antibody profiles to construct an age-seroprevalence relationship for antigenically variable pathogens.

[1] Oxford University Clinical Research Unit, Wellcome Trust Major Overseas Programme, Ho Chi Minh City, Vietnam. [2] Centre for Tropical Medicine and Global Health, Nuffield Department of Medicine, University of Oxford, Oxford, UK. [3] Center for Infectious Disease Dynamics, Department of Biology, Pennsylvania State University, University Park, PA, USA. [4] Department of Viroscience, Erasmus Medical Centre, Rotterdam, Netherlands. [5] Liverpool School of Tropical Medicine, Liverpool, UK. [6] Tropical and Infectious Disease Unit, Liverpool University Hospitals NHS Foundation Trust, Liverpool, England. [7] Hue Provincial Hospital, Hue, Thua Thien Hue Province, Vietnam. [8] Khanh Hoa Provincial Hospital, Nha Trang, Vietnam. [9] Dak Lak General Hospital, Buon Ma Thuot, Vietnam. [10] National Institute of Hygiene and Epidemiology, Hanoi, Vietnam. [11] Institute of Vector Borne Disease, Monash University, Melbourne, VIC, Australia. [12] Saw Swee Hock School of Public Health, National University of Singapore, Singapore, Singapore. [13] Hospital for Tropical Diseases, Ho Chi Minh City, Vietnam. ✉email: mfb9@psu.edu

The age–seroprevalence relationship is a basic epidemiological tool for understanding annual incidence and age-specific susceptibility of an infectious disease. There are two basic serological approaches for assessing the relationship between age and seroprevalence. Using long-term field studies, one can measure age-specific annual attack rates of a pathogen and infer what the resulting stable age–seroprevalence relationship should be based on the population's demographic parameters. Alternatively, using a single population cross-section, an age–seroprevalence curve can be inferred directly from the individuals' serological status, classified on a binary, discrete, or continuous scale. With both of these approaches, it is necessary to assume that exposure to the pathogen is constant in either time or age[1,2].

Multi-strain pathogens, however, present a challenge for the inference of age–seroprevalence relationships as infection with one strain typically triggers antibodies that cross-react against other strains. Strain-specific antibodies, like those binding to the host cell receptor binding domain of the influenza A virus particle, wane over time[3–5], potentially leading to underestimates of exposure when the estimates are based on assays that measure recent strain-specific antibodies. As a result, none of the single-strain age–seroprevalence curves presents an accurate history of pathogen circulation in a given population. For human influenza A virus, the existence of cross-reactions among different influenza strains or variants is well understood, as within-subtype cross-reactions among different strains are carefully characterized whenever a new strain emerges. An individual infected with an influenza strain in the year 2000 will have an antibody response that partially binds to or partially neutralizes (depending on the serological assay) influenza viruses circulating in 1995 or 2005. The strength of the cross-reaction wanes with increasing temporal distance between the strains, and it is known that antibodies to strains isolated closer together in time will cross-react more strongly (with some exceptions during longer periods of lineage co-circulation) than antibodies to strains isolated further apart in time[6–9]. A second important feature of influenza epidemiology and evolution that makes it challenging to understand age–seroprevalence relationships is that individuals of different ages will have been exposed to a different set of influenza strains. Older individuals will have been exposed to more strains than younger individuals, and some of these strains will have gone extinct before some of the younger individuals were born. Again, using a single influenza strain to generate an age–seroprevalence curve is not a solution to this problem, as only certain age bands of individuals will have been exposed to any particular strain. Indeed, age–seroprevalence relationships reported for influenza virus typically yield insight into the age-specific and time-specific patterns of infection of different strains and subtypes, but they do not have a monotonically increasing, saturating shape and cannot be used to estimate annual influenza seroincidence[10–14].

The rationale for constructing a general (i.e., not strain-specific) age–seroprevalence curve for influenza A virus is to infer long-run average attack rates, rather than the season-specific attack rates typically measured in cohort studies[13–16] and placebo arms in vaccine trials[17–22]. Serological studies performing inference on attack rates may also be limited by measurement errors[23], an inability to distinguish vaccinees from recently infected individuals, and an inability to distinguish individuals infected within the past year from those infected more than a year ago. Currently, the best methods for computing long-term attack rates of seasonal influenza are from large multi-strain serological analyses with inference on antibody responses, boosting, and waning[24,25], or exceptional data sets that present >10 years of surveillance[26,27].

Finally, in this study, we focus on influenza age–seroprevalence relationship in the tropics, as seasonal influenza attack rates are generally not known for tropical countries. One reason for the lack of measurement is an inability to identify a tropical influenza season[28–35] if one exists. Our study location is central and southern Vietnam. As annual influenza vaccine coverage in Vietnam is below 0.8% for our study period[36], the age–seroprevalence relationships presented here are the naturally accumulated age–antibody profiles in a population continually exposed to influenza virus with nearly no influenza vaccination.

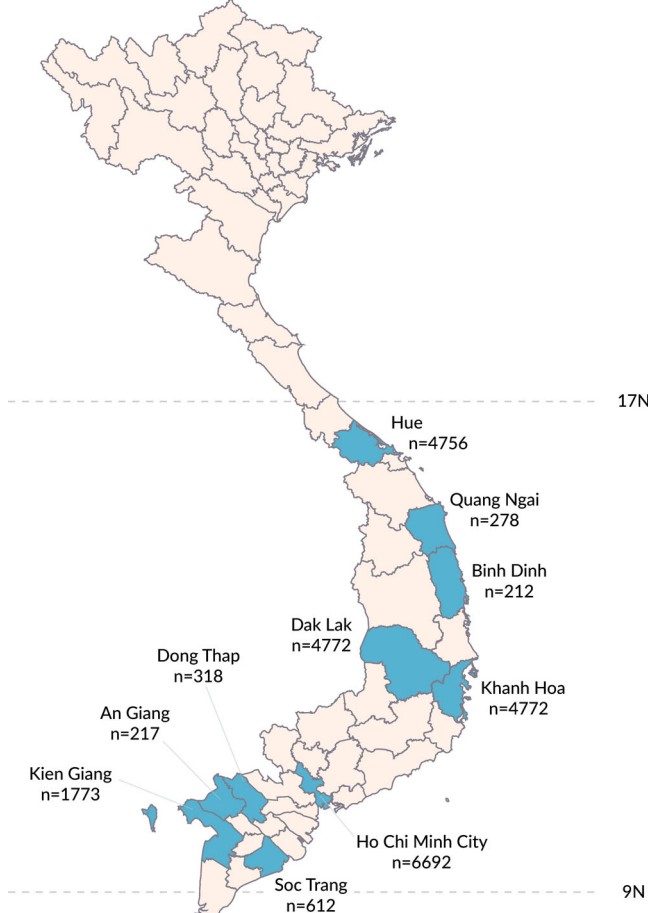

**Fig. 1 Serum collection sites at provincial hospitals in southern Vietnam that participated in this study.** Number of samples collected in each province is shown.

## Results

**Principal component structure of antibody profiles**. A principal component decomposition was conducted on 11 influenza antibody titers measured in 24,402 general population serum samples collected from 10 provinces in central and southern Vietnam (Fig. 1). This resulted in the first principal component (PC1) explaining 60.4% of the variance in the titer data and the second principal component (PC2) explaining an additional 16.5% of the variance. Figure 2A shows the 11 unit vectors $e_i$ corresponding to the 11 antigens in the assay projected onto PC1–PC2 space (all loadings shown in Supplementary Fig. 6). When the unit vectors $e_i$ in the original recentered 11-dimensional titer space are mapped to the basis vectors $v_i$ of the transformed PC space, the first coordinate (first principal component) of the $v_i$ is always positive, with a maximum 1.2-fold difference in magnitude among the 11 antigens, a consequence of a larger variance and range in H3 titers than in H1 titers. This also indicates that the first principal component is a positive-weighted sum of titers to

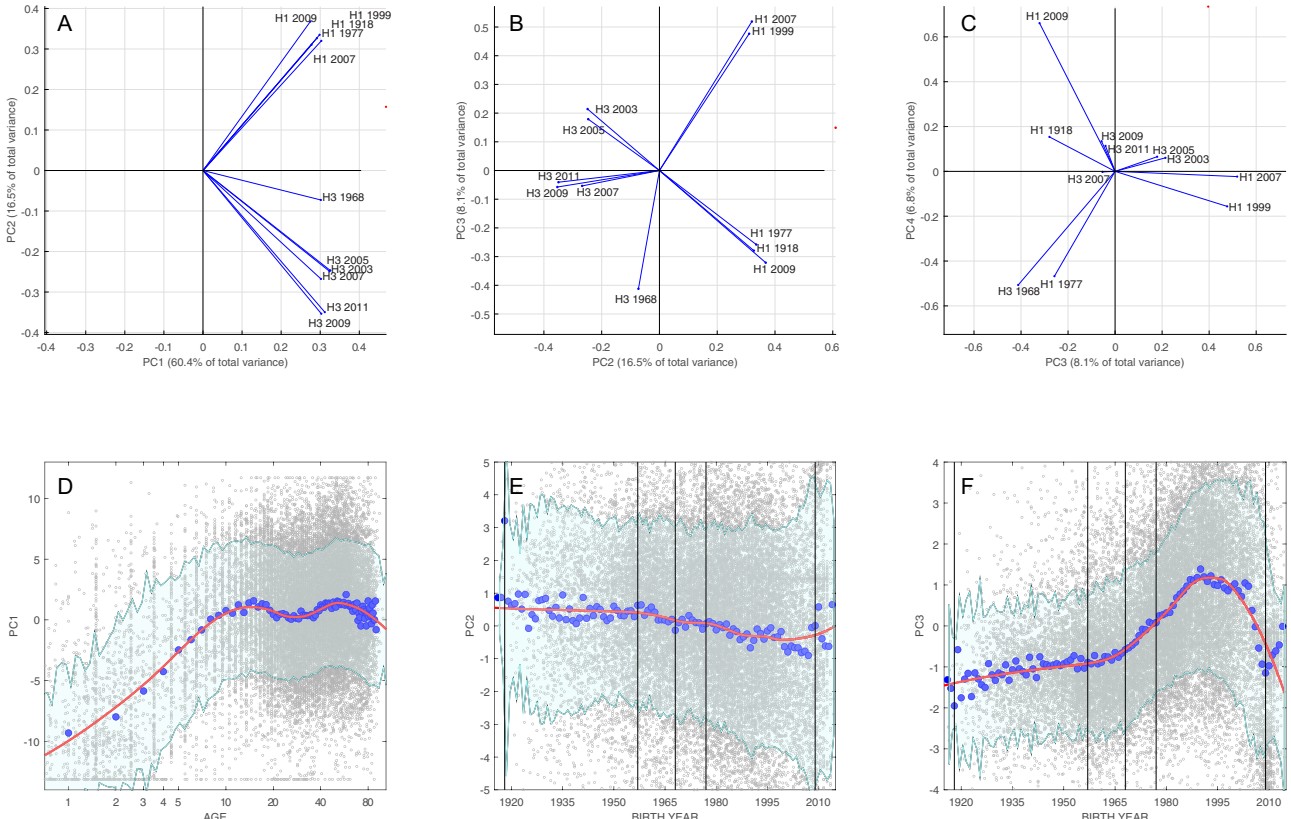

**Fig. 2 Principal component loadings and age/birth year relationships.** Principal component (PC) loadings for the first four principal components (**A–C**) show the PC coefficients of all 11 influenza antigens. Only two consecutive components are shown in each panel. **D–F** show the relationship between three first components and age (for PC1) or birth year (for PC2 and PC3). Small gray dots represent individuals, each with 11 titer measurements. The larger blue dots show the component mean for each 1-year age band or birth-year band. The red line is a spline regression curve of all 24,402 data points (LOESS curve, spanning factor = 0.5), and 80% prediction intervals (shown in green) were calculated using locally inferred error terms. The vertical lines show the time of introduction of new subtypes into the population. Note that titer scores were recentered around their means for this principal component decomposition and visualization, which is why the principal components (PC1, PC2, etc.) can be both positive and negative.

all antigens, suggesting that it can be used as a general measure of exposure and immunogenicity across all strains. We interpret PC1 as an indicator of composite antibody titer or seroprevalence in this analysis and note that as a continuous indicator it is more aptly viewed as a relative probability of exposure (or recent exposure) rather than a binary indicator of having been exposed or not. Although any positive-weighted sum of titer values can be assigned the meaning of "composite titer" or "total titer response" in a multi-strain epidemiological analysis, the derivation of PC1 in a principal component analysis (PCA) accounts for the fact that some antigens generate higher antibody titers than others, either because this is a property of the assay or because the viruses were truly more immunogenic in natural infections. The second coordinates (second principal component) of the basis vectors $v_i$ are positive exactly when $e_i$ corresponds to an H1N1 subtype and negative otherwise, indicating that the second principal component can be used to distinguish relative exposure to subtypes H1N1 and H3N2.

A serological age progression of the Vietnamese general population is shown in Fig. 3 on the first two principal component axes. The graphs are broken up into 1-year age bands through age 12 years and broader age categories thereafter, shown as density plots with darker colors indicating a higher density of individuals in a particular region of PC1–PC2 space. The PC1 axis corresponds to general exposure to influenza virus; note that PC1 values can be negative because all titer values are recentered around zero (i.e., they can be negative) during principal component decomposition. The PC2 axis shows relative exposure to H1N1 strains (positive

values) or H3N2 strains (negative values). The left-most points in the principal component space correspond to naive individuals (no H3 or H1 infection history) and the right-most points correspond to individuals that have maximum titers for all strains. The most striking feature of Fig. 3 is the consistent change observed in individuals in the early age classes, which shows influenza antibody acquisition in PC1–PC2 space for individuals aged 6 months to 12 years, even though this last age group may have lived through the circulation of three different H1 strains and five different H3 strains. In the early years of infection, individuals with H1 exposure only or H3 exposure only can be identified on the left edges of the diamond shape that makes up PC1–PC2 space (shown as a scatterplot in Supplementary Fig. 11).

The first principal component allows for a generalized way to describe current average antibody level to influenza A virus, without having to specify a particular strain or subtype (Fig. 2D). The largest change in PC1 can be seen in the first 10 years of life as children acquire their first influenza infection and generate a serological response. The value of PC1 appears to decline after age 10 (possibly an effect of original antigenic sin (OAS)[37–40]) and again for older individuals (>50 years), which may be a consequence of weaker immune/antibody responses in older individuals[41,42]. Differentiation in PC2 reveals some of the history of influenza pandemics in the twentieth century (Fig. 2E). Individuals born before 1957 ($n = 4054$) have the strongest titer responses to subtype H1 influenza, as do individuals born in 2009 or later ($n = 1580$), after the 2009 H1N1 pandemic virus emerged.

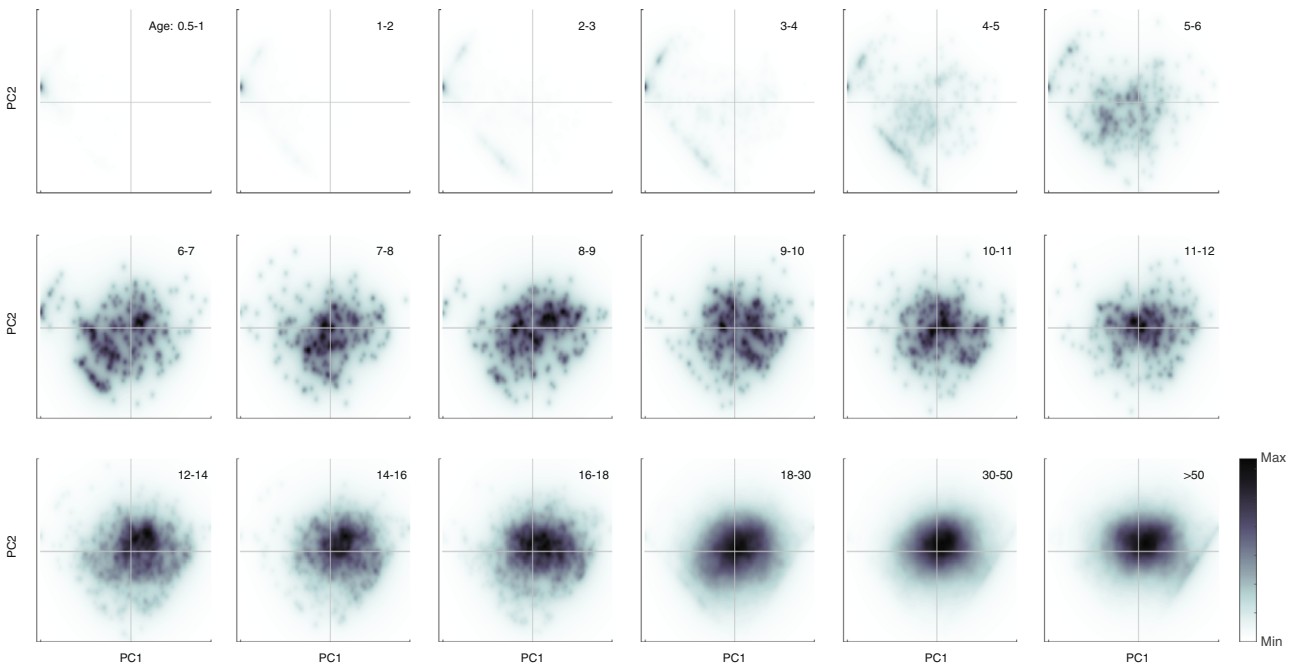

**Fig. 3 Density plots of individuals in principal component space, broken down by age group (shown in upper right of each panel).** Density is computed on a 256 × 256 grid spanning the minimum and maximum values of principal components one (PC1) and two (PC2). Color is scaled in each panel from zero ("Min," white) to the maximum number of individuals that appear in a pixel in that panel ("Max," black). Individuals move from the far left to the center and right of the PC1–PC2 space during the first 10 years of life, as a result of exposure to subtype H3N2 and H1N1 influenza viruses. The lower-left and upper-left boundaries of the diamond shape in PC1–PC2 space correspond to individuals who have only been exposed to one H1N1 strain (upper left) or those who have only been exposed to one H3N2 strain (lower left). Number of individuals (across all ages) shown here is $n = 24,402$.

Individuals born in the 1970s, 1980s, and 1990s generally have antibody titers that are more specific to H3N2 viruses (Fig. 2E), as these individuals were children when H3N2 subtypes were more prevalent. The higher H3N2 responses in this group probably reflect both (i) the generally higher antibody titers associated with H3N2 infections and (ii) OAS resulting from H3N2 infection in early childhood. Individuals born in the 1950s and 1960s are the most interesting with respect to order of subtype exposure as some would have experienced their early influenza exposures during the 1957–1967 gap when only H2N2 subtype viruses were circulating globally. None of the principal components distinguish this birth cohort particularly well; with no expected signal of OAS, these individuals' serological profiles likely represent a combined history of H1, H2, and H3 influenza infections.

The third principal component (PC3) breaks the H3N2 and H1N1 subtypes into further subgroups (Fig. 2B, C), which is also reflected in the age-related breakdown showing a strong contribution of H3 in the signal for those born between 1965 and 2009 (Fig. 2F), and the reverse during earlier and later time periods. For H3N2, the main qualitative feature of this structure is that responses to the 1968 H3N2 virus are unique—an expected outcome as these would only be seen strongly in individuals infected from 1968 to the early 1970s and should be independent from the occurrence of infection during the years 2003–2011. Subtype H3N2 viruses from 2007 to 2011 cluster together antigenically, and H3N2 viruses from 2003 and 2005 cluster together in a separate group. The third principal component also divides H1N1 viruses into two groups: one group that is antigenically similar to the 1918 Spanish influenza virus (H1-1918, H1-2009, and H1-1977) and a second group that represents the inter-pandemic circulation of H1N1 viruses in the 1990s and 2000s (Fig. 2B). The H1-1977 virus is the obvious outlier in the first group, and the fourth principal component distinguishes it from the 1918 and 2009 H1N1 strains (Fig. 2C).

In general, the principal components neatly sort individuals into birth cohorts (Supplementary Fig. 8), something that is not achieved by looking at antibody titers in isolation (Supplementary Fig. 7). Figure 4 shows histograms of maximum antibody titer by birth year (Fig. 4A), largest magnitude positive principal component by birth year (Fig. 4B), and largest magnitude negative principal component by birth year (Fig. 4C). PC1 is excluded from these analyses as it only corresponds to exposure without discriminating among antigens. If maximum magnitude is in PC2, this shows that an individual's antibody profile is dominated either by H1N1 exposures (if the component is positive and has largest absolute value, as in Fig. 4B) or H3N2 exposures (if the component is negative and has largest absolute value, as in Fig. 4C). As expected, individuals born after 2009 are likely to have an antibody profile dominated by a large and positive PC2, and individuals born between 1968 and 2009 are likely to have an antibody profile dominated by a large and negative PC2. Antibody profiles of individuals born in the 1980s and 1990s are dominated by a positive PC3, which corresponds to antibody specificity to H1-1999 and H1-2007. Individuals born after 2009 will also have a large and positive PC4 and a large and negative PC5; these correspond to exposure to H1-2009 and to H3-2009/H3-2011, respectively. Dominant antigen groupings are also readily visible in Fig. 4A, but these interpretations are susceptible to (i) the misreading of cross-reactions, e.g., individuals born after 2009 showing reactions to H1-1918, and (ii) confusing high antigen reactivity with infection history (e.g., individuals born in the 1970s and 1980s being dominated by an antibody response to H3-2003).

**Annual attack rates.** The clearest signal of infection history in our data comes in the form of PC1 correlating to the presence or absence of past influenza infection, allowing us to treat PC1 (a positive-weighted sum of antibody titers) as a proxy for seroprevalence. In general, for a titer vector ($\tau_1, \tau_2, ..., \tau_n$) of $n$ co-

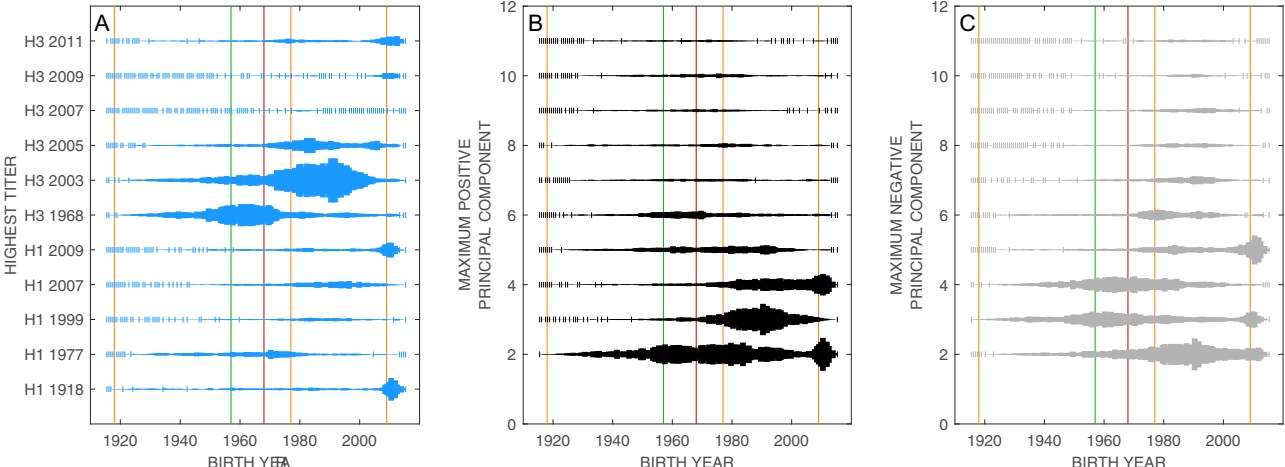

**Fig. 4 Histograms showing how individuals are sorted by titer or principal component.** Histograms showing how individuals are sorted by their **A** maximum antibody titer, **B** maximum positive principal component (excluding PC1), and **C** maximum negative principal component (excluding PC1). In each panel, each of the $n = 24,402$ individuals appears exactly once depending on their highest titer (**A**) or the maximum magnitude of their principal components (**B**, **C**). For example, most individuals born after 2009 have their maximum antibody titer to the 2011 H3N2 strain, the 2009 H1N1 pandemic virus, or the 1918 H1N1 influenza virus. These same individuals have either principal component 2 or 4 as their largest magnitude component (among positive components). The vertical lines show the time of introduction of new subtypes into the population: H1N1 (orange), H3N2 (red), and H2N2 (green).

circulating and non-cross-reacting pathogens, the first principal component will be proportional to the total number of pathogen exposures by age $a$ in a graph of PC1 against age. For a titer vector $(\tau_1, \tau_2, \ldots, \tau_n)$ of $n$ pathogens with high cross-reactivity, PC1 will be proportional to the probability of having been exposed to at least one of the $n$ pathogens. These are the two boundary situations, but for intermediate cross-reactivity there is no easily analogous interpretation, suggesting that the cross-reactivity parameter sigma $\sigma$ may play an outsized role in our ability to use multi-strain serological approaches for population-level inference of past pathogen exposure. In our data, the average cross-reactivity among neighboring H3 strains (strains that are 2 years apart) can be described by a standard Pearson correlation of $\sigma = 0.88$, which shows that within a subtype neighboring strains can be viewed as highly cross-reacting.

To assess the accuracy of using PC1 as a proxy for seroprevalence, we built an individual-based simulation of influenza infection, antibody acquisition, antibody waning, and back boosting[43] (Supplementary Methods). Comparing simulated attack rates to median PCA-inferred attack rates shows that the median inferred attack rates are within 5% of their simulated values (Supplementary Figs. 2 and 3); however, the stochastic nature of the simulation and the large variance in simulation outcomes does not allow us to conclude that our PCA-based estimator is unbiased. Crucially, there is no serologically measured long-term attack rate for influenza in Vietnam against which to validate our approach.

Separating the titer data by subtype to avoid having to account for low or intermediate cross-reactions between subtypes, we carry out two separate PCAs for H1 and H3 and infer the attack rates for each subtype separately (Supplementary Fig. 9). Optimizing the likelihood in Eq. (1), maximum-likelihood estimates for the annual attack rates in Vietnam are $AR_{H1} = 16.0\%$ (95% confidence interval (CI): 14.7–17.3%) and $AR_{H3} = 25.6\%$ (95% CI: 24.1–27.1%). The location-specific annual attack rates (Fig. 5) did not vary much from these estimates except for the attack rate of H1N1 in Hue, which was estimated at 21.6% (95% CI: 18.4–26.1%). There were no reports of Hue experiencing a larger pandemic wave[30] (or subsequent H1N1 epidemics) than other locations in Vietnam. However, variation in reporting patterns and asymptomatic/subclinical infection may occur[44], thus we cannot exclude the possibility that Hue experienced a higher rate of H1N1 infections than Khanh Hoa or Ho Chi Minh City.

## Discussion

An important challenge in serological analyses of antigenically variable pathogen families is the construction of a surrogate measure for seroprevalence that takes antibody cross-reactions and differential immunogenicity into account. The approach we propose here is dimensionality reduction across strains. By taking the first principal component of a data set of $n$ individuals whose antibodies have been measured to $m$ different antigens, we reduce an $m$-dimensional data set to a single dimension (PC1), which we use as a proxy for seroprevalence. With sufficient and representative age sampling in the younger age classes, the first principal component of such a data set should be a positive-weighted sum of an individual's antibody titers, with the weights adjusting for the higher titers or immunogenicities of some antigens when compared to others. The main challenge with this approach is validation of PC1 as a quantity that is in fact proportional to cumulative probability of exposure. In principle, this could be done with a cohort study run across enough influenza seasons to allow a long-term average attack rates to be measured. In practice, cohort studies like this are not common, and the task would be impossible for a data set of the size we present here. Despite this lack of direct validation with a known attack rate in the same locations and age groups, we can show that the long-term exposure patterns inferred using a PCA approach are (i) consistent with known descriptions of influenza epidemiology and influenza attack rates[15,24–26,45–47] and (ii) able to be validated with a simulation approach.

The estimates presented in this paper show that the average annual attack rate of influenza in Vietnam ranges from 13.4 to 21.6% for H1N1 and from 25.0 to 27.5% for H3N2. These ranges are consistent in their relative magnitudes, with H3N2 having a higher annual attack rate than H1N1[13], and in the implied age of first infection (ages 3–6 years) observed in cross-sectional data analyzed with traditional serology[48]. A second point of consistency is that peak PC1 value (i.e., peak weighted antibody titer) in the Vietnamese data occurs for individuals around age 10 years, approximately the age group predicted to have highest antibody titers resulting from OAS[40]. Our analysis was not designed to detect the effects of OAS, and the appearance of OAS in the PCA lends further credence to PC1 as an appropriate measure of composite titer or seroprevalence for influenza A virus.

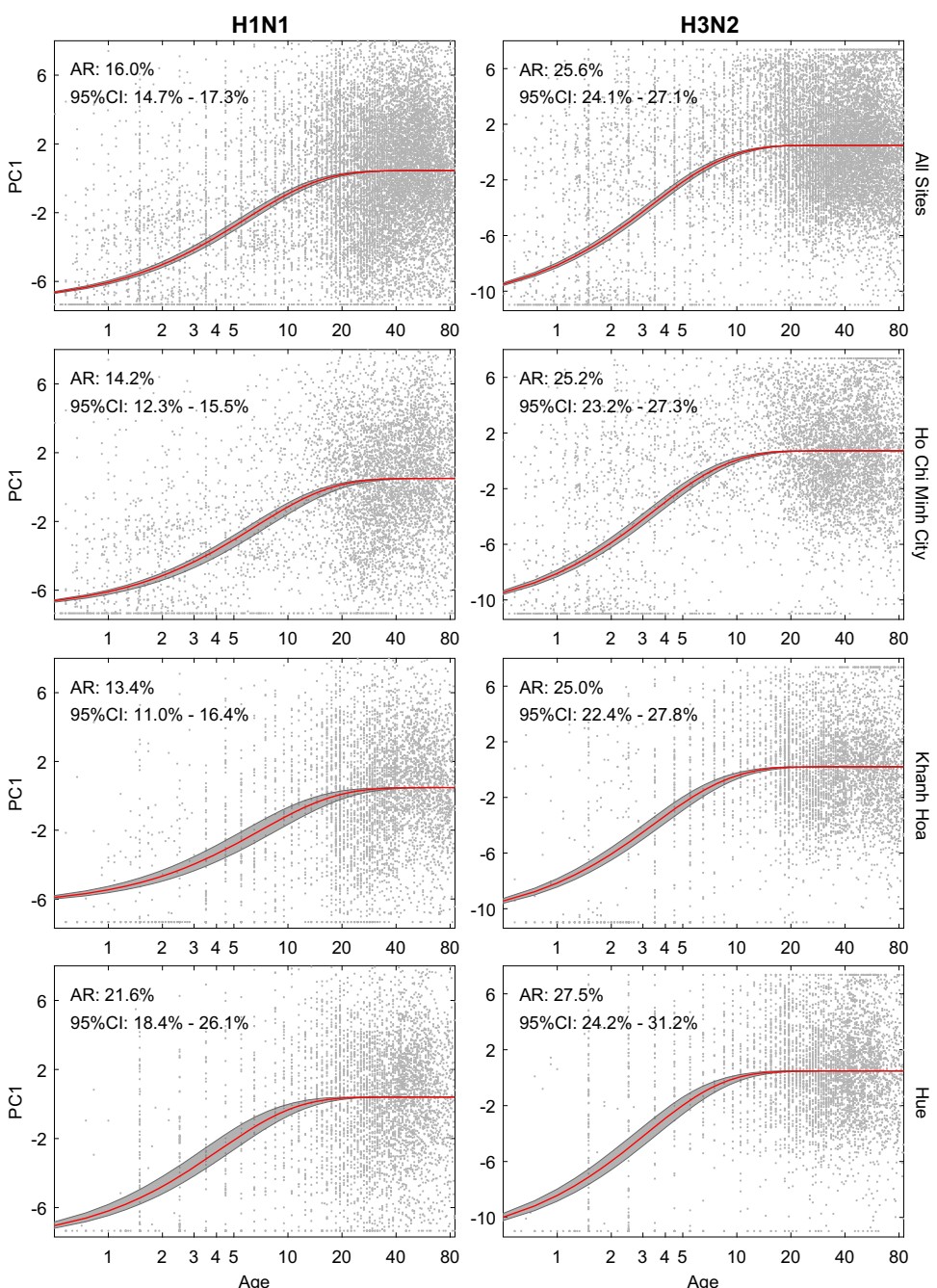

**Fig. 5 Age–seroprevalence curves shown by location and by subtype.** The four rows correspond to location, with the top row showing the age–seroprevalence relationship across Ho Chi Minh City, Hue, and Khanh Hoa combined ($n = 16,220$). The two columns show age–seroprevalence separately for subtypes H1N1 and H3N2 performed with separate PCAs. One gray dot is an individual. Red line shows the maximum-likelihood estimate of a saturating curve (Eq. 1) of the first principal component (PC1) as a function of age; gray bands show 95% confidence intervals. AR attack rate, CI confidence interval.

Despite the concordance with field data, further progress is needed on the theoretical justification and in silico validation of using PC1 as a marker of seroprevalence. The key theoretical question revolves around the interpretation of PC1 as it relates to the immune profile of an individual. Certainly, the neighbor-to-neighbor inter-strain cross-reactivity parameter $\sigma$ is influential in this interpretation as it affects whether an antibody profile represents a small number of past infections (high $\sigma$) or a large number of past infections (low $\sigma$). The next analytical step in this principal component approach will likely require creating a $\sigma$-adjusted PC1 so that it more accurately differentiates between the presence and the number of past

infections. An in silico validation approach could be used to test whether PC1 or a related construct is unbiased as an estimator of seroprevalence, but this simulation approach would itself need to be validated against field data. Specifically, the age distribution of influenza cases (symptomatic) is well known but the age distribution of infections (symptomatic and asymptomatic combined) is reported much less frequently. In addition, in a simulation, one needs to know how an antibody profile affects a person's likelihood of becoming an influenza infection, but clinical studies most commonly measure the effect of an antibody titer on becoming an influenza case. This measure may have to be estimated independently for

children and adults, as it is serological measurements in children that directly inform attack rate estimates. Individual-based simulation development would need to focus on accurate representations of titer profiles, protection from infection, and an accounting of all influenza infections across age groups.

As with any surrogate for seroprevalence, PC1 is limited in that it only provides distinguishing information for the younger age groups (here <10 years). Within these younger age groups, it is known that infection history and exposure do vary by age[49–53], but differences in single-year age bands are not always easy to estimate with a cross-sectional design (Fig. S12) unless large sample sizes are available. In our analysis, the teenage and adult age groups have similar values of PC1, indicating that most individuals aged >10 years are likely to have been exposed to influenza virus. In fact, the bottom six panels of Fig. 3 show that individuals' antibody profiles do not change much after an individual is likely to have been exposed to both H3N2 and H1N1 viruses; certain age groups will have high titers to viruses they were exposed to in childhood, but these exposure profiles are not qualitatively distinguishable from each other using PC1 alone. The utility of performing a PCA analysis on this general population serological data set from central and southern Vietnam lies in its location (tropical) and vaccination history (nearly none); reconstructing the natural age–seroprevalence relationship allows us to measure influenza A attack rates in a part of the world where influenza persists year-round and does not cause regular or predictable epidemics[28–30,54].

Moving beyond the first principal component, the remaining components give us an indication of serological/antigenic relatedness among strains (Fig. 2A–C). Although this relatedness is straightforward to characterize with sequencing and phylogenetic methods, a serological relatedness signal (more labor-intensive in its construction[6,8]) is more appropriately founded on the virus's phenotype rather than its genotype. As expected, these serological relatedness signals show that the 2003 and 2005 H3N2 strains cluster together; that the 2007, 2009, and 2011 H3N2 variants are related; and that the 1968 H3N2 is serologically distinct from H3N2 strains circulating after 2003. For the H1N1 subtypes, the 1999 H1N1 and 2007 H1N1 (both 1977 lineage) cluster with each other, while there are no obvious serological relationships among the 2009 H1N1 pandemic strain, the 1977 Russian flu, and the 1918 Spanish flu (these viruses cluster with each other in the first three principal components and are separated by the fourth). For the population, the primary utility of the higher principal components is the sorting of serological responses by birth cohort, which has already been found to influence vaccine efficacy[55–57], subsequent symptomatic infection[57–59], back boosting[60], and hospitalization and mortality[58]. An additional application of higher principal components may be in the construction of individual-level antibody profiles that could be used to assess susceptibility to influenza virus infection, as in Yang et al.[61].

The promise of large-scale serology is that certain traditional epidemiological variables—attack rate, age-specific susceptibility, cross-reactivity—will be able to be measured with higher precision. As large-scale multi-antigen serological approaches like this one become more common, it will become important to include field study components that allow for validation of results obtained from cross-sectional data alone. The present approach taken for influenza virus should be expanded to other well-characterized multi-strain pathogens such as dengue[62,63], norovirus[64–66], and pneumococcus[67,68] to share lessons on which inferential methods and study designs are most appropriate for the precise and robust estimation of a broad range of epidemiological quantities.

## Methods

**Data**. Serum samples used for this analysis come from a serum bank established in 2009 in southern Vietnam and maintained for the purposes of measuring exposure and seroincidence to a range of pathogens[36,69–74]. Every 2 months or every 4 months (depending on the site), 200 residual serum samples are collected from the hematology or biochemistry departments of ten major public hospitals: the Hospital for Tropical Diseases in Ho Chi Minh City, Hue Central Hospital in the city of Hue, Khanh Hoa General Hospital in Nha Trang, Dak Lak Provincial Hospital in Buon Ma Thuot city, Soc Trang General Hospital, Dong Thap General Hospital, Kien Giang General Hospital, Binh Dinh General Hospital, Quang Ngai General Hospital, An Giang General Hospital (Fig. 1). Approximately 7800 samples are collected each year and the samples are believed to represent the hospital-going population or general population in their respective provinces. For further details on sample collection, see Nhat et al.[36]. A total of 35,688 serum samples collected between 2009 and 2015 were selected for this analysis. The sample collection was approved by the Scientific and Ethical Committee of the Hospital for Tropical Diseases in Ho Chi Minh City and the Oxford Tropical Research Ethics Committee at the University of Oxford. All residual serum samples were collected anonymously from hospital laboratories with no identifiers included that could link the sample back to the original patient; there was no patient enrollment or consent procedure.

Samples were tested for IgG antibodies to the HA1 region of the influenza virus hemagglutinin protein for 11 different human influenza A viruses by a protein microarray[72,75,76], 5 strains of the H1 subtype and 6 strains of the H3 subtype (Table 1). Subtype H1 viruses included were the 1918 pandemic "Spanish Flu" virus (A/South Carolina/1/1918), the 1977 "Russian Flu" virus that was re-introduced into the population after a 20-year absence (A/USSR/92/1977), two antigenic variants of the 1977 lineage both of which were vaccine strains (A/New Caledonia/20/1999 and A/Brisbane/59/2007), and the 2009 "swine flu" pandemic virus (A/California/6/2009). Subtype H3 viruses included were the 1968 pandemic "Hong Kong" flu variant that re-introduced H3 circulation into human populations (A/Aichi/2/1968) and the five most recent H3N2 variants available at the time the study was designed: A/Wyoming/3/2003, A/Wisconsin/67/2005, A/Brisbane/10/2007, A/Victoria/210/2009, and A/Victoria/361/2011 (non-egg-adapted), three of which were vaccine strains.

Samples from individuals <6 months of age were excluded from the analysis to avoid assigning maternal antibody profiles to infants. The data set uses decimal ages based on information reported at the hospital; e.g., a 12-year-old is coded as 12.5 and a 6-week-old is coded as 0.12. After excluding individuals <6 months and samples that did not have computable titers for all 11 antigens (Nhat et al.[36], supplement, section 1.1), 11,286 samples were excluded for a total of 24,402 samples to be used in this analysis. Sample collection times are shown in Supplementary Fig. 5.

**Clustering**. Clustering of serological profiles was performed with a PCA (Matlab, R2019b, Mathworks, USA). PCA was performed on a $24{,}402 \times 11$ log-titer matrix with the Matlab function pca, which derives the principal component basis vectors through singular value decomposition. Titer values are all measured on the same scale, thus rescaling titer values was unnecessary. The first principal component was interpreted as a surrogate of seroprevalence (see "Results" section). To determine whether the magnitudes of the principal components corresponded to infection history, we sorted individuals by their largest magnitude (i.e., absolute value) principal component, looking at positive and negative components separately.

**Likelihood inference**. To estimate attack rates, we performed separate PCAs for three different locations (Ho Chi Minh City, Hue, Khanh Hoa) and the two influenza subtypes H3N2 and H1N1. Taking the first principal component (PC1) as a surrogate for weighted exposure to H3N2 or H1N1, we fit the PC1 value as a function of age to estimate the annual location-specific attack rates of subtypes H3N2 and H1N1 using the relationship

$$PC1(a) = H - Ke^{-\lambda a} \tag{1}$$

**Table 1 HA1 antigens of 11 different human influenza strains used for the study.**

| Antigen | Subtype | Abbreviation |
|---|---|---|
| A/South Carolina/1/1918 | H1N1 | H1-1918 |
| A/USSR/92/1977 | H1N1 | H1-1977 |
| A/New Caledonia/20/1999 | H1N1 | H1-1999 |
| A/Brisbane/59/2007 | H1N1 | H1-2007 |
| A/California/6/2009 | H1N1 | H1-2009 |
| A/Aichi/2/1968 | H3N2 | H3-1968 |
| A/Wyoming/3/2003 | H3N2 | H3-2003 |
| A/Wisconsin/67/2005 | H3N2 | H3-2005 |
| A/Brisbane/10/2007 | H3N2 | H3-2007 |
| A/Victoria/210/2009 | H3N2 | H3-2009 |
| A/Victoria/361/2011 | H3N2 | H3-2011 |

where $H$, $K$, and $\lambda$ are fitted parameters, with an individual's PC1 value counting as one data point. The $H$ and $K$ parameters are necessary to infer the minimum and saturating maximum of the PC1 curve as PC1 does not span the range zero to one as seroprevalence does. Optimization was done with a Nelder–Mead routine. The annual attack rate can be computed as follows

$$AR = 1 - e^{-\lambda} \qquad (2)$$

and 95% CIs were obtained through likelihood profiling. All analyses were conducted in Matlab. Attack rate analysis focused on three sites that had a sufficient number of samples from children aged <5 years: Ho Chi Minh City ($n = 1301$ samples from children <5 years), Khanh Hoa ($n = 358$), and Hue ($n = 383$).

**Validation.** To validate that PC1 is an appropriate surrogate for seroprevalence, an individual-based epidemic simulation was developed in C++ (Supplementary Methods). The simulation mimics the non-seasonal patterns of influenza A/H3N2 cases in Vietnam over a 10-year period[30,77] and uses influenza susceptibility data, antibody response data, and antibody waning data (measured with the same protein microarray) from a concurrently run clinical study run in Ho Chi Minh City from 2013 to 2015[78,79]. Simulations of 500,000 individuals are run with fixed attack rates ranging from 5 to 30%, and 6700 samples are taken at the exact sampling times (from 2010 to 2015) as in the serum collections for Ho Chi Minh City. A PCA is done on each simulated data set, and attack rate inference using Eq. (1) is done on the age and PC1 vectors.

**Reporting summary.** Further information on research design is available in the Nature Research Reporting Summary linked to this article.

## Data availability
Data are posted publicly[80] at https://doi.org/10.5281/zenodo.5594737.

## Code availability
Code is posted publicly[80] at https://doi.org/10.5281/zenodo.5594737.

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

## Acknowledgements

This work was funded by the Wellcome Trust Grant 089276/B/09/7 (D.N.V., N.T.D.T., P.H.A., G.E.T.) a Wellcome Trust Sir Henry Dale Fellowship and Enhancement Award (M.F.B., N.H.T.V., T.T.N.T., H.T.P., P.H.A., N.T.L.T., H.E.C.) Wellcome Trust Grant 097465/B/11/Z (S.T.) and by a British Medical Association HC Roscoe Award (2011–2014). C.P.S. is funded by the National Health and Medical Research Council of Australia. M.K. and E.d.B. are funded by Dutch Ministry of Economic Affairs, Agriculture, and Innovation, Castellum Project.

## Author contributions

D.N.V., M.K and M.F.B. conceptualized the study. D.N.V. and M.F.B. performed the analysis and wrote the paper. All authors read the paper and agreed to its contents. N.T.D.N. computed and validated titer measurements and oversaw the analysis pipeline from sample database to luminescence readout to titer value. E.d.B. printed microarrays and validated microarray performance. N.H.T.V., T.T.N.T., H.T.P. and P.H.A. performed the serological assays. T.M.Q. analyzed luminescence readouts that were input into titer calculations. N.T.L.T. managed the sample collection process. N.T.N.L., N.T.H.H. and T.T.K.H. collected samples. P.Q.T. and M.C. provided influenza incidence data in Vietnam and advised on individual-based simulation validation method. T.D.N. and M.F.B. wrote the individual-based simulation that was used in the validation procedure. S.T. provided clinical data on the titer protection threshold and the antibody waning rate. M.F.B., C.P.S., G.E.T., H.E.C. and N.V.V.C. oversaw data collection and study management.

## Competing interests

The authors declare no competing interests.
