## [Peer Review File · Nature Communications]

REVIEWER COMMENTS

Reviewer #1 (Remarks to the Author):

While it is interesting to use an PCA approach to analyze serology data with testing on multiple strains. The methodology and the interpretation is not very clear. What is the mechanistic rationale for the PCA approach?

1) The interpretation of the PCA is unclear. For example, individuals with high boosting and fast waning can show a similar pattern in serology with individuals with low boosting and slow waning, but PCA cannot account for that.

2) How is the back-boost handled by this approach? In particular, individuals cannot be infected with strains that only exist before they are born, but they can still have antibodies due to cross-reaction from infections.

3) It is unclear why the PCA with equation 1 can be used to estimate attack rate? What is the rationale for this? And what is the meaning of annual attack rate of a particular % for H3N2 or H1N1, when every year has an epidemic of a difference size? Although it is difficult as suggested in the Discussion section, some simulation should be done to validate this analytic approach.

4) Is it generalisable for the estimated PCA and what is the out-of-sample performance? For example if you randomly select half of the data and estimated PCA, and then fit with another half of the data, do you get a similar estimate for the AR?

Reviewer #2 (Remarks to the Author):

This paper presents a way to combine the data from multiple antibody assays to estimate an age-seroprevalence curve for a multi-strain pathogen. The method is applied to a large cross-sectional data set from a non-vaccinated setting, and natural attack rates of influenza A are calculated for various subsets of this study population.

My review is focused on the PCA aspects of the paper, as this is my area of expertise.

PCA is an appropriate method for analysing correlated numerical variables such as the eleven antibody titers presented in this paper. It is an interesting and, to my knowledge, novel, application of PCA. The first principal component (PC1) is a weighted average of the eleven antibody titers, and can therefore be used as a proxy for seroprevalence. This is a common occurrence in PCA; for measurements of physical dimensions on similar objects, for example, PC1 can often be interpreted as an indication of "overall size" of the objects.

I suggest adding a little more detail about the PCA method. For example, were the PCs calculated by singular value decomposition (SVD) of the titer value matrix, or by eigen decomposition of the covariance matrix? And though I read between the lines that the antibody titer values were not scaled before PCA, but it might be worth stating this explicitly in the methods section.

Eq. 1 (p7): "we fit the PC1 value as a function of age". However, in equation 1 age-specific seroprevalence is fitted as a function of age (a); it is unclear to me what the parameters H , K and λ refer to, and which of these indicate the PC1 scores in Equation 1.

Overall, the analysis is technically sound. The results are well presented and justify the conclusions.

MINOR COMMENTS:

(p8) "The PC2 axis shows relative exposure to H1N1 strains (positive axis) or H3N2 strains (negative axis)." -- In brackets it should be "(positive values)" and "(negative values)", respectively. There are no positive and negative axes.

The manuscript has many avoidable spelling mistakes. Some specific examples are given below.

Abstract:

"remaining pricipal components" -- principal

Methods - Data:

"Scientific and Ethical Committee" -- Scientific

Results - PC Structure of Antibody Profiles:

"corrosponds to an H1N1 subtype" -- corresponds

"first prinicipal component" -- principal

Results - Annual Attack Rates

"past influeza infection" - influenza

Discussion:

"validation of PC1 as a quanntity" -- quantity

"with H3N2 having a higher annual attach rate" -- attack

"seroprevalence for influeza A virus" -- influenza

"will be able to measured" -- revise this phrase

"mutli-antigen serological approaches" -- multi-antigen

Theo Pepler

Reviewer #3 (Remarks to the Author):

Introduction

1. para 2: where the authors discuss antibodies clustering in time, there are a few exceptions to this. For example, some of the currently-circulating H3 viruses are antigenically distinct; .e.g antisera raised against 3c3.A viruses does not inhibit 3c2.A viruses. Also, currently antigenic evolution is probably going to be quite limited and subject to bottlenecks.

2. p6: the eleven viruses used in the array should be described in the main text. Even with older viruses, some contemporaneous viruses were distinct or were known to have properties that limited interpretation but traditional serology methods.

3. If samples were collected until 2015, why are there no antigens for the years 2010-2015? These will be antigenically distinct and heterogeneous and should be included as well for the approach used to be of any use in the future. Virus diversity is increasing for both H3 and H1 viruses, so the utility of PCA under such diversity needs to be determined.

4. The naming of viruses using short of long names should be consistent. e.g. text on page 9 uses H1-18 but the corresponding figure used "H1 2018".

5. p13: the sorting of serological responses by birth cohort was also shown by Fonville (DOI: 10.1126/science.1256427) in their antibody landscapes paper, yet that seminal paper is not cited. There seem to be many parallels with the landscape approach and the approach proposed here, and it is an oversight that the authors haven't attempted to interpret their method in light of the antibody landscapes approach and where each method may align or diverge in interpretation. Further studies along these lines have been done to understand immunogenicity by birth cohort e.g. DOI: 10.1038/s41467-020-18465-x, <https://doi.org/10.1093/infdis/jiy376>, <https://doi.org/10.1093/infdis/jiz201>, <https://doi.org/10.1371/journal.ppat.1008109>

6. It is a little unclear how the specificity of PCA enables inference about attack rates and the authors should be clear about whether they expect these inferences to be possible for a single season (without paired sera straddling the season), or for longer periods of antigenic evolution.

7. Figure 5: it would be helpful to use the same y scale to highlight that ARs were higher for H3.

Dear Referees

Thank you for these helpful remarks. Apologies for the long reply letter below, but our one substantial change requires a lot of explanation.

The referees are right that it's better to validate this PC1-as-seroprevalence interpretation with a simulation approach. We have done this. The challenge here is that there is no validated piece of software that is the gold-standard influenza simulation. This may at first sound like a nit-picky critique (after all, many simulations have been built and they do seem to recover the basics of influenza epidemiology), but the truth is that there are some important features of these simulations that have not been validated.

First, what is the age distribution of influenza infections? We know what the age distribution of cases is from surveillance and clinical studies, but we don't know what the age distribution is for all infections when we count asymptomatic infections, sub-clinical infections, and symptomatic cases together. You can get the age-distribution of infections from cohort studies that have pre- and post-epidemic serum samples, but these are specific to the particular years and locations where the cohort was run, have small sample sizes, and are not generalizable. And you can get the age distribution of cumulative infection (through cross sectional study) but not the age distribution of infections in a particular year or epidemic.

Second, what is the protective antibody titer for an *infection*, not a *case*? An HI titer of 40 or a microarray titer of 100 gives 50% protection from becoming a symptomatic influenza case (in a test-negative case-control design). But we don't know what this protective titer is for simply becoming infected or not. An individual with an HI titer of 40 may become infected, but they may have a low viral load and few symptoms, and thus not be recorded as a case.

Why does this matter? To validate, we have to build a simulation where influenza infections (not cases) occur as they do in the real world. We know that an individual with high antibody titer is unlikely to become a case, but in the simulation we need to know if this person is likely or unlikely to become infected at all (and thus develop antibodies or not). Alternatively, we could simply validate our simulated age-distribution of infections (where age is a rough proxy for titer in the early age classes) with a real-world age-distribution of infections, but this data set does not exist.

Nevertheless, taking all of these limitations into account, we have built an individual-based simulation for this validation purpose. We have made some reasonable assumptions about infection based on current titer. And we have described how sensitive the PC1-based inference system is to these assumptions. This is all detailed in the supplement and in the responses below.

Best Wishes

Dao Nguyen Vinh, Maciej Boni, on behalf of all authors

Reviewer #1 (Remarks to the Author):

While it is interesting to use an PCA approach to analyze serology data with testing on multiple strains. The methodology and the interpretation is not very clear. What is the mechanistic rationale for the PCA approach?

The two basic rationales are

1. that PCA shows how similar individuals are in their (multi-dimensional) antibody profiles. And yes, as the referee notes below, this cannot account for the fact that a high-titer fast-waning individual will look the same as a low-titer slow-waning individual.
2. that the first principal component (PC1) essentially orders individuals along an axis of low titers to high titers (because PC1 is a positive-weighted sum of titers), and allows us to test if PC1 is an appropriate surrogate for seroprevalence in a multi-strain pathogen systems.

More details below on this topic in comments below.

1) The interpretation of the PCA is unclear. For example, individuals with high boosting and fast waning can show a similar pattern in serology with individuals with low boosting and slow waning, but PCA cannot account for that.

Yes, the referee is right here, and as far as we can see there is no workaround for this. On an individual scale there is no way to distinguish these two individuals. This is also true for standard antibody titer measurements (when not using principal components).

However, on a population scale we can compare 2-year-olds to 3-year-olds, or individuals born in 1950 to those born in 1970. The immune system is not so unpredictable that these waning/boosting differences would cause us to conflate all 2-year-olds with all 3-year-olds due to lack of differentiability in their antibody profiles. And in fact these two age groups are differentiable. Specifically, for the attack-rate analysis, they are differentiable in these two key quantities: their average titer and the percentage of individuals with some positive titers.

Likewise, individuals born in the 1950s or 1960s are well past their fast-waning dynamics, and they can be characterized as, on average, having higher titers to H1 viruses or higher titers to H3 viruses, based on their birth year. Yes, there are some individuals born in 1955 who may randomly appear to have the antibody profile of an individual born in 1975. But, when averaging across many individuals, the birth cohorts do look different.

There are many new comments in the manuscript (in green) on the interpretation of the PCA approach.

2) How is the back-boost handled by this approach? In particular, individuals cannot be infected with strains that only exist before they are born, but they can still have antibodies due to cross-reaction from infections.

There are two questions above. How do you handle cross-reactions and how do you handle back boosts.

Cross reactions are not handled in any special way outside the PCA. Individuals with a high H3-2005 titer are also likely to have a similar H3-2003 titer, due to cross-reaction, even if they were born in January 2005 and could not have been infected with the H3-2003 strain. The PCA accounts for this as an individual with a high H3-2003 titer and an individual with a high H3-2005 titer are both mapped to the same values of PC1, PC2, PC3, PC4, etc. It is only in the last several principal components that some differentiation between H3-2003 and H3-2005 can be seen.

Back boosts were not considered in the originally submitted manuscript. In the new simulation work that we have done for the revision, we examine scenarios with and without back boosts. When back boosts are included, all past positive titers are boosted (by a factor of 7.26) and they wane back down to their original level after one year. The value of 7.26 was chosen because it matches the empirically measured antibody waning rate after one year. Boost values of 2, 5, and 8 were explored and they did not have much of an effect on the attack rate inference. These simulation outputs and graphs are included in the GitHub repository.

3) It is unclear why the PCA with equation 1 can be used to estimate attack rate? What is the rationale for this? And what is the meaning of annual attack rate of a particular % for H3N2 or H1N1, when every year has an epidemic of a difference size? Although it is difficult as suggested in the Discussion section, some simulation should be done to validate this analytic approach.

Yes, this seems to be the most important comment in the referee responses: that a simulation approach is needed to validate the approach of using PC1 as seroprevalence. We spent a couple months coding up an individual-based simulation for this purpose and performing the validation. The main challenge (as we found out) is that there is no single validated gold-standard influenza simulation. The one we developed replicates H3N2 influenza dynamics in Vietnam very well, but we cannot verify that the simulation's age-structure of infection in children matches what occurs in nature (because there is no data). Our simulation produces a lot of variation in infection numbers in children, and this has an outsized effect on attack-rate inference. The median PCA-inferred attack rates are close to the simulations' input attack rates, which is good news as a computational validation that PC1 can be used as a surrogate for seroprevalence. Nevertheless, more work will need to be done in this area to understand the true mathematical relationship between seroprevalence and PC1. We leave this for a future theoretical paper, and we describe some of the challenges in paragraph 3 of the discussion.

The rationale for using PC1 as a surrogate for seroprevalence is that a weighted sum of titers (i.e. PC1) is an appropriate surrogate for whether someone has ever been infected or has never been infected, simply

because a completely naïve individual should have no detectable titers. The construction of PC1 accounts for the fact that some titers tend to be higher than others (or more easily detectable than others).

Yes, the attack rate is different every year. And it can be measured in cohort studies (easy) and sometimes in cross-sectional studies (difficult in Vietnam due to lack of seasonality). What we are measuring with our PCA-as-seroprevalence approach is the long-run average attack rate over ten years or more. Essentially, the antibody titers of individuals in the 0-10 age group determine the average attack rate over the past ten years. This is stated (we think, clearly) on page 4 (green text).

4) Is it generalisable for the estimated PCA and what is the out-of-sample performance? For example if you randomly select half of the data and estimated PCA, and then fit with another half of the data, do you get a similar estimate for the AR?

Excellent question, thank you. This has been done and is shown in Figure S4. With a sample size of 24,402 the structure of PC space is very stable when taking subsets of the data. As the subsets get smaller, the number of children in each subset gets smaller, and this does have an effect on the attack rate inference which is primarily determined by the PC1 values of younger children.

Reviewer #2 (Remarks to the Author):

This paper presents a way to combine the data from multiple antibody assays to estimate an age-seroprevalence curve for a multi-strain pathogen. The method is applied to a large cross-sectional data set from a non-vaccinated setting, and natural attack rates of influenza A are calculated for various subsets of this study population.

My review is focused on the PCA aspects of the paper, as this is my area of expertise.

PCA is an appropriate method for analysing correlated numerical variables such as the eleven antibody titers presented in this paper. It is an interesting and, to my knowledge, novel, application of PCA. The first principal component (PC1) is a weighted average of the eleven antibody titers, and can therefore be used as a proxy for seroprevalence. This is a common occurrence in PCA; for measurements of physical dimensions on similar objects, for example, PC1 can often be interpreted as an indication of "overall size" of the objects.

Thank you for this positive feedback.

I suggest adding a little more detail about the PCA method. For example, were the PCs calculated by singular value decomposition (SVD) of the titer value matrix, or by eigen decomposition of the covariance matrix? And though I read between the lines that the antibody titer values were not scaled before PCA, but it might be worth stating this explicitly in the methods section.

Singular value decomposition of the titer matrix. Antibody titers were logged but not scaled, as they are already all on the same scale, from a minimum of zero to a maximum of 7.5. This is now stated on page 5.

Eq. 1 (p7): "we fit the PC1 value as a function of age". However, in equation 1 age-specific seroprevalence is fitted as a function of age (a); it is unclear to me what the parameters H, K and lambda refer to, and which of these indicate the PC1 scores in Equation 1.

H and K are simply the inferred max and min of the PC1 curve which is concave and saturates at a maximum value H. This is now made clear on page 6.

Overall, the analysis is technically sound. The results are well presented and justify the conclusions.

Thank you.

MINOR COMMENTS:

(p8) "The PC2 axis shows relative exposure to H1N1 strains (positive axis) or H3N2 strains (negative axis)." -- In brackets it should be "(positive values)" and "(negative values)", respectively. There are no positive and negative axes.

Amended as above.

The manuscript has many avoidable spelling mistakes. Some specific examples are given below.

Abstract:

"remaining pricipal components" -- principal

Methods - Data:

"Scientific and Ethical Committee" -- Scientific

Results - PC Structure of Antibody Profiles:

"corrosponds to an H1N1 subtype" -- corresponds

"first prinicipal component" -- principal

Results - Annual Attack Rates

"past influeza infection" - influenza

Discussion:

"validation of PC1 as a quantity" -- quantity

"with H3N2 having a higher annual attach rate" -- attack

"seroprevalence for influeza A virus" -- influenza

"will be able to measured" -- revise this phrase
"mutli-antigen serological approaches" -- multi-antigen

Sorry about this. Spell check was turned off. Thank you! All above has been fixed (and more).

Reviewer #3 (Remarks to the Author):

Introduction

1. para 2: where the authors discuss antibodies clustering in time, there are a few exceptions to this. For example, some of the currently-circulating H3 viruses are antigenically distinct; .e.g antisera raised against 3c3.A viruses does not inhibit 3c2.A viruses. Also, currently antigenic evolution is probably going to be quite limited and subject to bottlenecks.

Yes. A concise review of this is available in Allen & Ross (2018) [doi: 10.1080/21645515.2018.1462639], and the Nelson et al (2006-2008) papers show similar lineage co-circulation on local levels in the United States.

We have added a parenthetical caveat stating this in para 2 of the introduction. However, if the editors and referees agree, we would prefer to take this out as it breaks up the sentences and flow in this paragraph. The purpose of this paragraph is to give the reader a brief introduction (refresher) to the largely unidirectional antigenic evolution of human influenza viruses. While it's true that during longer periods of lineage co-circulation the 'closer in time, closer antigenically' correlation does not always hold, this is something that isn't needed to understand our PC1 approach to measuring seroprevalence.

2. p6: the eleven viruses used in the array should be described in the main text. Even with older viruses, some contemporaneous viruses were distinct or were known to have properties that limited interpretation [by] traditional serology methods.

Yes, we have added a few short sentences introducing these viruses in the methods section.

There are sometimes issues with influenza viruses changing their agglutination properties as the virus evolves (thus affecting interpretation of HI assays) as well as changes in whether cell culture or egg adaptation is used for a vaccine strain (and this can affect interpretation of serology and protection). We don't use HI assays in our work, and we mention now that we use the original non-egg-adapted 2011 Victoria strain; in other words the viral antigen in our assay is matched to the virus the population would have experienced in natural infection. Since our assays measure binding to the HA1 protein (and not agglutination of RBCs) we can't say for certain whether there are any virus-specific issues affecting our assays.

3. If samples were collected until 2015, why are there no antigens for the years 2010-2015? These will be antigenically distinct and heterogeneous and should be included as well for the approach used to be of any

use in the future. Virus diversity is increasing for both H3 and H1 viruses, so the utility of PCA under such diversity needs to be determined.

This is simply because the study was designed in 2010 and 2011. The microarrays had to be printed, which took more than a year. In 2014, we did add the Switzerland 2013 H3N2 variant to the microarrays to update them, but only a small proportion of samples were tested with these newer arrays with the extra antigen, so we have left these data out in our analysis.

A study with additional diversity (including strains from 2013, 2015, and on) would be terrific, but it is beyond what we can do at the moment.

In fact, this was one of the main drawbacks of our large-scale approach, that with a small team it took several years to print and validate the microarrays, process samples, analyze the data, etc.

4. The naming of viruses using short of long names should be consistent. e.g. text on page 9 uses H1-18 but the corresponding figure used "H1 2018".

Yes, thank you. The naming is now uniform.

5. p13: the sorting of serological responses by birth cohort was also shown by Fonville (DOI: 10.1126/science.1256427) in their antibody landscapes paper, yet that seminal paper is not cited. There seem to be many parallels with the landscape approach and the approach proposed here, and it is an oversight that the authors haven't attempted to interpret their method in light of the antibody landscapes approach and where each method may align or diverge in interpretation. Further studies along these lines have been done to understand immunogenicity by birth cohort e.g. DOI: 10.1038/s41467-020-18465-x, <https://doi.org/10.1093/infdis/jiy376>, <https://doi.org/10.1093/infdis/jiz201>, <https://doi.org/10.1371/journal.ppat.1008109>

Yes, thank you. Certainly, we are very familiar with the Fonville work since this was based out of a study at OUCRU Vietnam. The key difference between the Fonville (2014), Derek Smith (2004), and other papers out of this group is that they perform multi-dimensional scaling on the virus phenotype, whereas we perform multi-dimensional scaling on a human antibody profiles. These are related of course. But, their question is 'how similar are these two viruses?' while our question is 'how similar are these two people?'.

Some of the priming work cited by the referee is very relevant to our work (thank you for the nudge). The Liu paper shows clear evidence of a back-boost that is modulated by the antigenic cluster an individual was primed with (for H1N1). The Gostic paper shows that NA-imprinting may affect one's probability of becoming a 'case' if infected with the same NA subtype. The Budd paper shows a similar effect to the Gostic paper (including mortality and hospitalization effects) but there are no statistical equations included in this paper so it is a little harder to understand the effects, including how age was factored out.

We have added comments into our discussion (page 13, green text) on known effects of priming, sorting into birth cohorts, and back boosting. In addition, in our new simulation work we included the Fonville back-boost feature, in the way it was observed for most individuals in the paper, as a check to see if back-boosting had an effect on our validation exercise (Figure S3).

6. It is a little unclear how the specificity of PCA enables inference about attack rates and the authors should be clear about whether they expect these inferences to be possible for a single season (without paired sera straddling the season), or for longer periods of antigenic evolution.

It is for long-run attack rate. Not for a single season. Explained (clearly we think) on page 4.

7. Figure 5: it would be helpful to use the same y scale to highlight that ARs were higher for H3.

The y-scale here is an arbitrary PC1 value, and it is a different PC1 for H1 and for H3, so even if they were on the same scale they would not be comparable.

REVIEWER COMMENTS

Reviewer #1 (Remarks to the Author):

Thanks for the authors' intensive efforts on addressing my comments, this version is much clearer than before. I still have some follow-up comments and suggestions to assist the readers understand and interpret the authors' work. Overall, while using PCA is an interesting approach, I did not see how this improved the methodology on analyzing serology data for influenza. For example, Kucharski PLoS Bio 2018 developed a more comprehensive model by including those boosting and waning patterns in the model, and also did not assume the annual attack rates are the same over years.

Specific comments:

1. I think the authors are estimating the "long-run average attack rates", I am unclear on the epidemiological importance. I would rather think it is a major limitation that this assumption is used to simplify the analysis, as we know the attack rate for each season is different for many reasons. Please clarify if it is indeed a reasonable assumption.

2. (Follow-up of Q3 in last review) I am still unclear on why equation 1 could be used to estimate the "annual attack rate".

My understanding is that you fit the estimated PC1 value to get the parameters, i.e. $PC1 = H - K * \exp(-\lambda * a)$, and get the value of H, K and λ ? if so, I still did not get why such an approach would get the annual attack rates by $1 - \exp(-\lambda)$. In the PCA the boosting and waning pattern, the cross-reaction and protection from HAI titers is not implicitly accounted for. Please state if there is any mechanistic rationale for this approach. For example, is this equation derived based on some population transmission model? If not, the simulation would be particularly important to validate this.

3. Thanks for the authors' intensive effort for simulations. I have some comments and suggestions about the simulations.

a. The authors discussed several components in the simulation that may impact the results, which I agree. However, as the PCA approach did not consider this, it is important to validate that the PCR approach is robust to this assumption. The authors tried different values of beta (the protection of HAI titers), but children and adults could also be added. If the protection of HAI titer of the age distribution of cases (i.e the susceptibility difference between children and adults) could impact the PCA approach, some discussions on the impact and the potential solution is needed.

b. 10 replications are difficult to interpret if the approach can recover the true value of attack rates. This should be increased to at least 50. The number of confidence intervals that cover the true value in those replications should be presented.

4. The authors stated that one contribution of the paper is to propose the use of PCA to do dimension reduction. However, in Yang PLoS Pathogen et al, (<https://journals.plos.org/plospathogens/article/authors?id=10.1371/journal.ppat.1008635>), they also proposed easy to use metrics to do the dimension reduction. The similarity and difference should be discussed. What are the advantages of using PCA given that it is more complicated?

Reviewer #3 (Remarks to the Author):

The authors have done a thorough job of responding to the comments of the reviewers.

Apologies for the delay in re-submission. We finished the new simulations in mid-March, but have been extremely busy over the past few weeks. Please find our responses to the new comments below.

Best Wishes

Dao Nguyen Vinh, Maciej Boni, on behalf of all authors

Reviewer #1 (Remarks to the Author):

Thanks for the authors' intensive efforts on addressing my comments, this version is much clearer than before. I still have some follow-up comments and suggestions to assist the readers understand and interpret the authors' work. Overall, while using PCA is an interesting approach, I did not see how this improved the methodology on analyzing serology data for influenza. For example, Kucharski PLoS Bio 2018 developed a more comprehensive model by including those boosting and waning patterns in the model, and also did not assume the annual attack rates are the same over years.

Yes, the Kucharski 2018 paper uses cohorts of 151 participants (Guangdong, China) and 69 participants (Ha Nam, Vietnam) to look at individual-level waning data. They also obtain attack rate estimates from the Vietnam cohort. These cohorts span 2007-2012 for Vietnam (total starting participants around 1000) and 2009-forward for China (thousands of starting participants, if I remember correctly). Note that attack rates are presented for the Vietnam data only, because these data are collected longitudinally, i.e. they have multiple serum samples per person. In the China sample set, only one sample per person is available, so Kucharski et al are not able to provide an attack-rate estimate.

We are not aiming to improve the Kucharski methodology. We are introducing a methodology for cross-sectional data. The Kucharski 2018 Vietnam analysis looks at longitudinal data. When you have longitudinal data, you can easily check to see who was infected between 2007 and 2008, between 2008 and 2009, etc. because you have the names and addresses of all the participants, and you can follow everyone up every year to see if the attack rate in 2010 was different than the attack rate in 2011 or 2012. Horby et al 2012 (*Am J Epid*, vol 175, p1062) present the attack rates for the Ha Nam data, and the Kucharski 2018 analysis refines these estimates.

(cross-sectional studies are much less expensive than longitudinal studies, which is why it's important to have methods for extracting reliable attack rates from cross-sectional studies).

Specific comments:

1. I think the authors are estimating the "long-run average attack rates", I am unclear on the epidemiological importance. I would rather think it is a major limitation that this assumption is used to simplify the analysis, as we know the attack rate for each season is different for many reasons. Please clarify if it is indeed a reasonable assumption.

This is a standard approach in classic epidemiology. In modern epidemiology, a good introduction to this topic is in Grenfell and Anderson 1986 (*Journal of Hygiene*, vol 95, p419), and the bottom panel of their Figure 1 shows the same type of age-seroprevalence curve that we are constructing in our work. If you look below their equation (2) on page 421, you will see their in-text equation " $y(a) = 1 - x(a)$ " or if you substitute equation (2) the equation is " $y(a) = 1 - \exp(-\lambda a)$ ", which is the same as our equation (1) when our $H=1$ and $K=1$. We added the H and K parameters because our scale is "value of the first principal component" and not "fraction of individuals not yet infected." Grenfell and Anderson can assume H and K are both equal to one, but we are required to fit them. This approach is

now included in most modern epidemiology textbooks, and it is usually called a ‘catalytic model’. This approach was used earlier, in the 1940s and 1950s, when antibody assays became more common as scientific tools. But, the 1986 reference above is a good starting reference for modern application.

So, the λ in the Grenfell equation and the λ in our equation gets you the “average attack rate”. λ is a per-year rate of infection, so the actual attack rate (or, average fraction of the population that gets infected per year) is $1 - e^{-\lambda}$.

If your data set contains individuals aged 0 to 10, then λ gives you the average attack rate over the last 10 years. This is why we call it the “long-run average attack rate.” It is the average attack rate, over the past ten years, for a virus that has reached an equilibrium-level of transmission with its host population.

We would like to remind the referee and the editors that our attack-rate estimate is calculated in the known absence of influenza vaccination. Therefore, we are sure that all of seropositive individuals have antibodies generated by infection, not antibodies generated by vaccination. These types of estimates are not possible in temperate-zone influenza data sets where annually >20% of the population is vaccinated.

2. (Follow-up of Q3 in last review) I am still unclear on why equation 1 could be used to estimate the "annual attack rate".

My understanding is that you fit the estimated PC1 value to get the parameters, i.e. $PC1 = H - K * \exp(-\lambda * a)$, and get the value of H, K and lambda?

Yes, correct.

if so, I still did not get why such an approach would get the annual attack rates by $1 - \exp(-\lambda)$. In the PCA the boosting and waning pattern, the cross-reaction and protection from HAI titers is not implicitly accounted for. Please state if there is any mechanistic rationale for this approach. For example, is this equation derived based on some population transmission model? If not, the simulation would be particularly important to validate this.

The Grenfell and Anderson paper and explanation above is the rationale for why “ $1 - \exp(-\lambda)$ ” is a common approach to estimating an annual attack rate.

Another explanation, in a discrete formulation, is to look at

p = probability of being infected during a one year period

and the probability of being ‘never infected’ is $(1-p)$ for a 1-year old, $(1-p)^3$ for a 3-year old, $(1-p)^{10}$ for a 10-year-old, etc. The above is a geometric distribution. The exponential distribution in the Grenfell paper is simply a continuous version of a geometric distribution. The approaches are completely equivalent. The investigator just needs to choose if they want to work with discrete data (3 year-olds, 5 year-olds) or continuous data (3.2 years-olds, 5.4 year-olds, etc.).

Just to clarify for the referees and editors, the PCA cannot take boosting or waning into consideration. The PCA simply operates on a static data set (completely agnostic of how the data set was generated) and it re-organizes or re-orientes the data points on a different set of axes so that variation can be viewed in an easier and more convenient way. The simulation exercise in the appendix *does check* if the PCA approach is affected by boosting patterns or antibody waning in the simulated output, but boosting and waning cannot be integrated into the PCA analysis itself.

3. Thanks for the authors' intensive effort for simulations. I have some comments and suggestions about the simulations.

a. The authors discussed several components in the simulation that may impact the results, which I agree. However, as the PCA approach did not consider this, it is important to validate that the PCA approach is robust to this assumption.

We believe the referee here is referring to the (i) the degree of protection from becoming an influenza case versus infection, (ii) the age distribution of infections in an influenza epidemic, (iii) the cross-reactivity parameter between neighboring influenza strains, and (iv) the fixed-attack rate approach in the simulation we developed in Nov/Dec 2020. These are listed in section 1.1 in the supplement.

Again, to clarify, the PCA cannot take any of these mechanisms into consideration. The PCA simply operates on a static data set (completely agnostic of how the data set was generated) and re-organizes the data points on a different set of axes so that variation can be viewed in an easier and more convenient way.

The referee is correct that it is important to test robustness. We tested different values of β to look at the effects of (i) and (ii) above. We do not need to test for the effects of differing cross-reactivity since we have a large data set and we can readily estimate the cross-reactivity σ between neighboring influenza strains.

We would like to test the effects of different simulation types (point (iv) above) but this is somewhat beyond the scope of the current paper. The individual-based simulation we developed in Nov/Dec 2020 would normally be something that we would do for a stand-alone publication, as this is very labor-intensive and challenging to maintain and expand; we brought in three new co-authors and leveraged a data set that is not yet published to ensure that our simulation and data were both based on recent influenza epidemiology in Vietnam. Exploring the range of possibilities from different simulation types is something that is best left to a publication focused on individual-based epidemic simulation of influenza.

The authors tried different values of beta (the protection of HAI titers), but children and adults could also be added. If the protection of HAI titer of the age distribution of cases (i.e the susceptibility difference between children and adults) could impact the PCA approach, some discussions on the impact and the potential solution is needed.

Yes, our simulation could include differences in the way children and adults are protected by a particular antibody titer. We don't have data on this, but it is an interesting possibility to consider. Adding in these new simulation features and obtaining the data to see if this effect is a real is something that is best left to a study dedicated to this question. We have added a discussion point on this (in green) as suggested by the referee.

b. 10 replications are difficult to interpret if the approach can recover the true value of attack rates. This should be increased to at least 50. The number of confidence intervals that cover the true value in those replications should be presented.

Yes, the referee is correct here. This took a little bit of time, but we increased these simulations numbers to 50. Individual-based simulation is computationally costly. We have to run our simulation for 36 years to achieve equilibrium, and the large population size means that it takes many days to generate a new validation figure (assuming no bugs and no other hiccups).

Our data set is very large and thus the confidence intervals on the attack rates are very narrow. This is not a good measure of whether the simulated and inferred values coincide. It's the simulation output itself that is variable, not the sampling from the simulation output. This is why we are presenting inter-quartile ranges (IQR) in these figures,

so that these IQRs cover a large range of the simulation outputs and show how often in a ‘very random’ simulation you have agreement between the simulated and inferred outputs.

4. The authors stated that one contribution of the paper is to propose the use of PCA to do dimension reduction. However, in Yang PLoS Pathogen et al, (<https://journals.plos.org/plospathogens/article/authors?id=10.1371/journal.ppat.1008635>), they also proposed easy to use metrics to do the dimension reduction. The similarity and difference should be discussed. What are the advantages of using PCA given that it is more complicated?

Yes, the metrics in Yang et al are terrific. We have used these on the Todd (2016) study data and found them very useful. And like in our paper, the Yang metrics are developed per person, as opposed to per virus (as in the Kucharski papers, Fonville et al, Derek Smith’s papers, and others). The metrics in Yang et al are not used to construct an age-seroprevalence profile, rather, they are used to assess susceptibility per person. Nevertheless, this is a useful parallel. We have added a comment in the discussion, in green text.

(this paper was overlooked initially because it was published in July 2020, and we submitted our paper in August 2020).

Reviewer #3 (Remarks to the Author):

The authors have done a thorough job of responding to the comments of the reviewers.

Thank you for the refereeing effort and comments.

REVIEWER COMMENTS

Reviewer #1 (Remarks to the Author):

No further comments

Reviewer #4 (Remarks to the Author):

Infection with one influenza strain can generate cross-reactive responses to others, making interpretation of serological difficult. There are two main analysis approaches for dealing with this complexity: mechanistic approaches (i.e. develop a model of latent infections and antibody dynamics, then use data to estimate frequency of infection in different groups, as in Kucharski et al, PLOS Bio 2015), or statistical approaches (i.e. use large dataset to identify immunological predictors of infection history and apply these to estimate infection dynamics). In this paper, the authors use the latter approach, converting serological panel data into principal components, and using these components to predict average attack rates over time.

I appreciate that the manuscript has already gone through a round of review, and my comments therefore focus on the methodological aspects outstanding.

Main comments:

- The authors include a 'semi-mechanistic' step, applying a catalytic model to estimate the force of infection λ (which has a clear epidemiological interpretation) from PC1. This makes interpretation of the method difficult, because PC1 doesn't have a straightforward epidemiological meaning such as "proportion previously infected". Selecting this catalytic process as a monotonically increasing function for the model isn't necessarily inappropriate if the method can generate robust predictions (indeed a lot of machine learning algorithms have high predictive power but little mechanistic insights). However, the analysis presented in the supplement isn't convincing as identifying a 'surrogate of seroprevalence' - rather, it validates that the overall analysis pipeline can correctly recover the 'true' simulated attack rate in the presence of cross-reactive antibody dynamics. For example, there is nothing that shows how $PC1=X$ can be converted to $seroprevalence=Y$. So it would be helpful to frame the validation exercise and methodology accordingly.
- The authors assume a fixed force of infection across age groups, but there is much evidence that influenza varies with age (e.g. Cohen et al, Lancet GH, 2021). This could be estimated directly e.g. by calculating the gradient of $\log(PC1)$ vs age to get the rate of change in PC1 for each year of life (equivalent to λ if the gradient is constant).
- I found Figure 1 hard to follow. In the Results, it states PC1 values are positive, but in Fig 1D, they appear to span negative values as well?

Thank you very much to Referee #4 for the additional feedback. Our comments are below in blue. Revision 3 of our manuscript has new or modified text in green.

Please forgive the substantial delay in returning these revision. I was travelling all of June, and we had many students leave this summer, causing delays in some project.

Maciej Boni

Reviewer #4 (Remarks to the Author):

Infection with one influenza strain can generate cross-reactive responses to others, making interpretation of serological difficult. There are two main analysis approaches for dealing with this complexity: mechanistic approaches (i.e. develop a model of latent infections and antibody dynamics, then use data to estimate frequency of infection in different groups, as in Kucharski et al, PLOS Bio 2015), or statistical approaches (i.e. use large dataset to identify immunological predictors of infection history and apply these to estimate infection dynamics). In this paper, the authors use the latter approach, converting serological panel data into principal components, and using these components to predict average attack rates over time.

I appreciate that the manuscript has already gone through a round of review, and my comments therefore focus on the methodological aspects outstanding.

Main comments:

- The authors include a 'semi-mechanistic' step, applying a catalytic model to estimate the force of infection λ (which has a clear epidemiological interpretation) from PC1. This makes interpretation of the method difficult, because PC1 doesn't have a straightforward epidemiological meaning such as "proportion previously infected".

Yes, exactly. This is the main challenge in this type of work. We cannot have a measure of "proportion previously infected" for all of influenza since different strains circulate at different times, so we propose a surrogate measure, PC1.

After all the analysis has been done, is there a straightforward epidemiological meaning to PC1? Yes, PC1 is a weighted sum of titers, so it can be viewed as the 'total antibody response across all strains'. This will be higher in individuals with more exposures and lower in individuals with fewer exposures. We could have replaced the PC1 vector with ones and zeros based on whether a particular titer was above a threshold

or below a threshold; this would have given us number of exposures by age, and allowed us to compute the attack rate. We chose not to do this in order to not lose information from the titer measurements themselves.

PC1 is a weighted sum because it accounts for the fact that some strains are more immunogenic (or that the assay is more sensitive for some strains).

This is our interpretation of PC1 (presented in Results paragraph 1). Of course, as noted below by the referee, this needs some validation.

Selecting this catalytic process as a monotonically increasing function for the model isn't necessarily inappropriate if the method can generate robust predictions (indeed a lot of machine learning algorithms have high predictive power but little mechanistic insights). However, the analysis presented in the supplement isn't convincing as identifying a 'surrogate of seroprevalence' - rather, it validates that the overall analysis pipeline can correctly recover the 'true' simulated attack rate in the presence of cross-reactive antibody dynamics.

We understand the referee's point of view here.

As the referee points out, we have validated (via an individual-based simulation approach) that inference on PC1 can recover the simulated attack rate. Normally, you would want to use a seroprevalence estimate to obtain the true attack rate, but for influenza this is not possible. It is not possible because the multi-strain structure of influenza virus means we have many seroprevalence measurements instead of just one.

PC1 is a close correlate of the "traditional seroprevalence" curve, the curve that tells you the probability that someone in age group a has been infected in the past. The validation exercise in section 1 of the supplement demonstrates this. Because PC1 is a correlate seroprevalence, we call it a surrogate in the text. We could change this terminology to 'correlate' if the referee and editors prefer this.

The referee is correct that the validation exercise provides no mechanism or rationale for *why* PC1 should be a good surrogate/correlate for seroprevalence, or *how* we should interpret PC1 when viewing it as a measure of attack rate or seroprevalence. The mechanism and interpretation are described in the blue block of text starting at the bottom of page 1 of this document. We reiterate some of these ideas below.

We propose one mechanistic interpretation in paragraph 1 of the Results section, namely, that PC1 is a positive-weighted sum of titers and that it can be viewed as the total antibody response across all strains. This weighted sum will be higher with higher total lifetime exposure – this is the mechanism linking a human's reality of being exposed to multiple influenza viruses and this same human's PC1 value in the analysis. (Antibody waning and post-infection titer values need to be accounted for here, and these features are included in the individual-based simulation used in the validation approach).

There may be other interpretations. Our proposed interpretation may not be the definitive answer of how PC1 relates to seroprevalence. We believe this is still an open question mainly because the relationship

between σ (the cross-reactivity) and PC1 likely influences the relationship between PC1 and seroprevalence (this is our current opinion). This question is better left to a mathematically-focused paper on principal components analysis, and a study of whether a relationship can be derived between PC1 and the number of past exposures if the value of σ is known. We have discussed this with a mathematician colleague. This mathematical line of research is not within the scope of the current data-centered paper, so we have decided to separate this out as an independent project.

We do discuss this new challenge briefly in the manuscript (Results para 6, Discussion para 3) where we show the reader how a titer vector or a titer profile $(\tau_1, \tau_2, \dots, \tau_n)$ will be viewed differently if σ is high versus if σ is low. These comments were in our original submitted manuscript Aug 21 2020, and we have clarified them in this round of revision to make some of the links and interpretations more direct.

For example, there is nothing that shows how PC1=X can be converted to seroprevalence=Y. So it would be helpful to frame the validation exercise and methodology accordingly.

There is no direct conversion from PC1 to seroprevalence for a general data set. The principal components decomposition must be done first on a specific data set, then the force-of-infection parameter λ must be fit to the age-PC1 curve. After this is done, there is a direct conversion between PC1 and force of infection,

$$PC1(a) = H - K e^{-\lambda a}$$

where λ , H , and K are parameters that are fit from the data. In other words, λ is an estimator based on the PC1 values, i.e.

$$\hat{\lambda}(\{PC1_1, PC1_2, \dots, PC1_n\})$$

Once you have the force of infection estimate $\hat{\lambda}$, you can compute the expected seroprevalence by age.

We realize that we casually wrote equation (1) with “seroprevalence” rather than PC1 on the left-hand side. This has been changed to make things clearer.

- The authors assume a fixed force of infection across age groups, but there is much evidence that influenza varies with age (e.g. Cohen et al, Lancet GH, 2021). This could be estimated directly e.g. by calculating the gradient of $\log(PC1)$ vs age to get the rate of change in PC1 for each year of life (equivalent to lambda if the gradient is constant).

Yes, influenza infection does vary by age, and there are a number of cohort, school-based, and early-stage pandemic studies that touch on this topic. In our data, this effect is visible for H3N2 (using the approach above) if we look at all children under age 10. If we look at all children under age 5, this effect is more difficult to see. If we look at all children under age 7, this effect is weakly present. We have added a figure on this (Figure S12) and the figure caption expresses the necessary uncertainty.

We have also added a comment in para 4 of the discussion. We caution not to interpret these results too strongly as our data set and study type were not designed to answer age-based susceptibility questions. The best studies for this question have some information on contact rates or titer levels in children before infection, and this is how they are able to assess age-specific susceptibility.

- I found Figure 2 hard to follow. In the Results, it states PC1 values are positive, but in Fig 2D, they appear to span negative values as well?

Yes, this is because in a PCA analysis, the values are typically re-centered around their means so that visualizations like in Figure 2A, 2B, 2C are possible. This is standard for PCA presentation, and we have decided to keep the presentation this way. We could undo the centering and revert to the original coordinates. All of the y-axes in our figures would then be relabeled with the new PC values, but we would lose the ability to visualize the PC vectors in the top row of Figure 2.

We have clarified this for the reader in a number of places in the text (para 2 of results, Figure 2 caption).

(A reminder here that Referee #2 had specific expertise in PCA and stated that the PCA analysis in our manuscript is “technically sound” and “well presented”.)

REVIEWERS' COMMENTS

Reviewer #4 (Remarks to the Author):

The authors have addressed my previous comments.

One remaining thing that would be worth doing is to improve the documentation in the accompanying code (<https://github.com/bonilab/seroepi-02FL-influenza-vietnam-PCA>) - at a minimum, I would suggest documenting what the different files do and providing a brief 'quick start' script so readers can easily compile and run.

Reviewer #4 (Remarks to the Author):

The authors have addressed my previous comments.

One remaining thing that would be worth doing is to improve the documentation in the accompanying code (<https://github.com/bonilab/seroepi-02FL-influenza-vietnam-PCA>) - at a minimum, I would suggest documenting what the different files do and providing a brief 'quick start' script so readers can easily compile and run.

Yes, very good idea. We have added two README.md files in the matlab code and C++ code directories. In addition, we have improved the commenting of the code itself and pointed the reader to several relevant code sections that they might want to look at.

Rather than provide a quick start script, we simply named all of the matlab scripts in order as script 1 ("S01_Basic_PCA_Analysis.m), script 2 (S02_PCA_HeatMaps.m), etc. This way, the interested reader can simply load the scripts in order into matlab and press run on each one. This is all described in the readme file for the matlab code folder.